# Cognitive Effects of *Toxoplasma* and CMV Infections: A Cross-Sectional Study of 557 Young Adults Considering Modulation by Sex and Rh Factor

**DOI:** 10.3390/pathogens13050363

**Published:** 2024-04-28

**Authors:** Jaroslav Flegr, Veronika Chvátalová, Lenka Příplatová, Petr Tureček, Petr Kodym, Blanka Šebánková, Šárka Kaňková

**Affiliations:** 1Laboratory of Evolutionary Biology, Department of Philosophy and History of Sciences, Faculty of Science, Charles University, Viničná 7, 128 00 Prague, Czech Republicpetr.turecek@natur.cuni.cz (P.T.); blanka.sebankova@natur.cuni.cz (B.Š.); sarka.kankova@natur.cuni.cz (Š.K.); 2National Reference Laboratory for Toxoplasmosis, Šrobárova 48, 100 42 Prague, Czech Republic

**Keywords:** Rh factor, toxoplasmosis, cytomegalovirus, cognition, intelligence, memory, psychomotor performance, parasite, behavior, manipulation hypothesis

## Abstract

One-third of humanity harbors a lifelong infection with *Toxoplasma gondii*, and probably about 80% are infected with human cytomegalovirus (CMV). This study aims to delineate the associations between toxoplasmosis and cognitive abilities and compare these to the associations with CMV. We evaluated the cognitive performance of 557 students, who had been examined for *Toxoplasma* and CMV infections, using intelligence, memory, and psychomotor tests. The results indicated cognitive impairments in seropositive individuals for both pathogens, with variations in cognitive impact related to sex and the Rh factor. Specifically, *Toxoplasma* infection was associated with lower IQ in men, whereas CMV was predominantly associated with worse performance by women when testing memory and reaction speeds. Analysis of the antibody concentrations indicated that certain *Toxoplasma*-associated cognitive detrimental effects may wane (impaired intelligence) or worsen (impaired reaction times) over time following infection. The findings imply that the cognitive impairments caused by both neurotropic pathogens are likely due to pathological changes in the brain rather than from direct manipulative action by the parasites.

## 1. Introduction

*Toxoplasma gondii* is a protozoan parasite that undergoes sexual reproduction in its definitive hosts, namely, any species of feline. A broad spectrum of homoiotherm vertebrates, including humans, serve as intermediate hosts, wherein *Toxoplasma* reproduces asexually. Approximately one-third of the global human population is infected with *Toxoplasma* [1], as shown in Figure 1.

Historically, latent *Toxoplasma* infection was considered asymptomatic. However, research over the past two decades indicates that individuals with latent toxoplasmosis exhibit a higher incidence of various diseases and generally score lower on health-related metrics compared to those without *Toxoplasma* [5]; for a review, see [6]. *Toxoplasma* has also been demonstrated to alter the behavior of its intermediate hosts. Examples of such behavioral alterations include a shift from a natural aversion to cat odors to attraction being exhibited in laboratory-infected rodents [7,8], as well as prolonged reaction times in both infected rodents [9] and humans [10]. These changes have been hypothesized as being the result of manipulations by the parasite to increase the likelihood of transmission to its definitive host, a cat, by predation. An alternative and likely simpler explanation is that these differences are merely side effects of the host’s deteriorated health. The side effects hypothesis was challenged by the results of path analysis in a recent study [11], showing that deteriorated health does not mediate the observed differences in psychological traits. It is crucial to note that this study specifically tested and refuted the mediating role of physical health solely for the psychological changes examined, not in the context of cognitive impairment.

Compared to *Toxoplasma*, cytomegalovirus (CMV), a member of the herpesvirus family, infects an even larger proportion of the human population. The seroprevalence of CMV correlates with population age, being around 50% among university students and exceeding 80% in older demographics [12]. While latent postnatal CMV infection, akin to toxoplasmosis, was traditionally viewed as asymptomatic, emerging studies hint at its association with specific behavioral traits [13,14,15,16,17,18,19]. However, the behavioral impact of latent CMV infection has not received as much investigative attention as latent toxoplasmosis. Unlike toxoplasmosis, to date, there are no known behavioral experiments involving artificial CMV infection in laboratory animals, and longitudinal human studies are scarce [20,21]. Therefore, the causality relationship between infection and behavioral changes remains unresolved in the case of CMV.

Notably, CMV transmission does not occur through predation but instead through direct physical contact or bodily fluid exchange (e.g., during kissing). Based on the assumptions of evolutionary parasitology, this should result in less specific and less conspicuous expected behavioral modifications. This subtlety perhaps contributes to the topic’s lesser appeal for biologists who are interested in parasite manipulation. As of January 2024, the Web of Science database included 142 articles identified with the search query ‘toxoplasm* AND behavio*’, with 117 of these specifically addressing behavioral changes in latent toxoplasmosis. In contrast, the search query ‘(CMV OR cytomegalo*) AND behavio*’ yielded only 35 articles, and just 4 of these focused on postnatal CMV infection and its associated behavioral symptoms.

During the past 20 years of research focused on the association between latent toxoplasmosis and human behavior and cognitive performance, it has been revealed that the Rh factor of the infected individual plays a significant role as a modifier of these associations. Approximately 16% of Europeans are Rh-negative [22], a condition characterized by the absence of the immunodominant D epitope on the erythrocyte’s transmembrane protein-glycoprotein complex. The complex plays a role in the transport of ions across the erythrocyte membrane; however, its biological functions have not been fully elucidated [23,24,25,26,27].

There is evidence that Rh-negative individuals may experience various health challenges, including cognitive and neurological issues [28,29,30], which are probably related to localized hypoxia and heightened neuroinflammation in specific brain regions [30,31].

Numerous studies have underscored the finding that individuals with the Rh-positive phenotype, particularly heterozygotes with one D and one d allele of the RHD gene, exhibited increased resilience against negative influences like fatigue, smoking, and aging [32,33,34,35]. Intriguingly, this resilience also extends to *Toxoplasma* infection. Latent toxoplasmosis has been shown to have different impacts on the behavior, performance, and health of individuals with Rh-negative and Rh-positive blood types [32]. For example, uninfected Rh-negative individuals usually display quicker reaction times compared to uninfected Rh-positive individuals. However, upon *Toxoplasma* infection, Rh-negative individuals often experience a substantial decline in reaction times. In contrast, Rh-positive heterozygotes tend to maintain or even improve their reaction times after infection [32,35]. The underlying mechanisms of this *Toxoplasma*–Rh interaction are still not understood, yet its existence has been corroborated by approximately 10 studies over the last 15 years. This interaction also exhibits sex-specific variations; for more details, see [30]. No analogous phenomena have been noted in the context of human cytomegalovirus infection.

The general objective of our current study is to investigate the relationship of *Toxoplasma* or CMV infections with cognitive performance in a cohort of 557 university students. Both pathogens are capable of surviving for decades in the latent stages within the brain tissues of infected immunocompetent humans, where they can cause histopathological changes, including localized inflammatory lesions [36,37]. Unlike *Toxoplasma*, which is transmitted trophically and may, thus, gain an advantage by reducing the vigilance and cognitive abilities of its intermediate host, CMV is not suspected of engaging in a similar adaptive manipulation of host behavior and cognition. Therefore, by comparing the cognitive performance deficits associated with *Toxoplasma* and CMV infections, we aim to discern whether these are solely the result of pathological changes in the brain or if they are also the outcome of deliberate manipulative activity by the pathogen to facilitate its transmission from infected to uninfected hosts. For this purpose, we utilized a comprehensive suite of tests assessing intelligence, knowledge, memory, simple reaction times, and information processing speed. Additionally, in the exploratory part of the study, we investigated how sex and the Rh factor, recognized for their influence on toxoplasmosis-associated behavioral symptoms, impact both infections. Specifically, for toxoplasmosis, we investigated how sex and the Rh factor modulate specific aspects of cognitive performance. For CMV, the potential modulation of cognitive performance by sex and the Rh factor had not previously been studied; thus, our primary aim was to determine whether these factors influence the association between infection and cognitive performance.

## 2. Method

### 2.1. Subjects

We invited all our undergraduate students enrolled in Evolutionary Biology and Practical Methodology of Science courses from 2010 to 2015 to participate voluntarily in a research project investigating the relationship between parasites and human behavior, performance, and personality. The invitation was extended to the students during lectures and about 50% of the students consented, subsequently visiting our lab to provide 5 mL of blood for serological analysis, which was collected by trained medical personnel. Owing to the extensive nature of the cognitive and reaction tests, which required approximately 11 h per participant, our intention to keep the examination conditions identical in all rounds, and the limited capacity of our testing facility (a maximum of 12 subjects simultaneously), the testing was conducted over an extended period. Intelligence, knowledge, and declarative memory assessments were carried out between 2 and 24 months post-blood collection. An additional 1–6 months later, the participants returned for a third session to evaluate other memory types and reaction times. In total, datasets were collected from 557 students. During cognitive performance testing, neither the subjects nor the researchers were aware of the participants’ CMV and *Toxoplasma* infection statuses. Both male and female subjects were included in the study. Sex was identified based on the participant’s response to the demographic questionnaire query: ‘Your official sex at birth’. The study adhered to the Declaration of Helsinki guidelines and received approval from the IRB of the Faculty of Science at Charles University (protocol numbers 2009/06 and 2013/07). All participants were adults who provided written informed consent, as approved by the IRB.

### 2.2. Cognitive Performance Testing

#### 2.2.1. Intelligence

For the intelligence assessment, we utilized the Czech version of the Intelligence Structure Test I-S-T 2000 R [38,39]. This test is divided into two main modules: the basic module and the memory module. The basic module consists of three verbal, three numerical, and three abstract figural reasoning subtests. The memory module includes two memory tests and a knowledge test, which covers topics from geography and history, business, arts and culture, mathematics, science, and daily life. Employing both modules allowed us to comprehensively evaluate verbal, numerical, and figural intelligence, as well as verbal and figural memory and verbal, numerical, and figural knowledge. These source variables were then used to calculate fluid intelligence, crystallized intelligence, general intelligence, and general knowledge (for further details, see the Memory subsection). The participants’ intelligence, memory, and knowledge scores were derived from raw variables (i.e., the number of correct answers) using normative tables established for the entire Czech population. These tables do not account for age and sex; however, we controlled for these variables in our subsequent statistical analyses (refer to the Data Analysis section for details).

The I-S-T 2000 R test was administered on computers to groups of 7–12 individuals simultaneously in the same room, with all sessions starting at 9:15 a.m. The total duration of the test was approximately 250 min, including a 15-min break before the second module (comprising memory and knowledge tests). In conjunction with the I-S-T 2000 R session, the participants also engaged in 2 physical performance tests, provided saliva samples for hormone concentration analysis, and underwent ECG recording. These additional data were utilized in unrelated studies.

#### 2.2.2. Memory

For the assessment of associative memory performance, we employed the I-S-T 2000 R memory module. The first part, the verbal memory test, involved a 60-second learning phase where participants were presented with 13 words across 5 categories, such as food, sports, and professions, each starting with a unique letter. Subsequently, they were asked to identify the category of a word beginning with a specified letter, such as ‘B’ (10 questions, with a maximum raw score of 10). The second part, the figural memory test, required participants to learn 13 pairs of abstract figures within 60 s. They were then shown 13 figures and given 3 min to match each figure with its pair from 5 options (13 questions, with a maximum raw score of 13). Associative memory was determined by summing correct answers from both tests, leading to a maximum raw score of 23. Due to technical issues, we were not able to administer the second part of the I-S-T 2000 R, which contains memory tests, to approximately one-third of the participants.

In the third testing session, we utilized the three-part Meili memory test [40] to evaluate other memory types, namely spatial memory, free recall memory, and recognition memory. Participants initially had 60 s to memorize 30 objects in a 6 × 5 grid of simple black and white drawings, such as a car, hat, candle, and flower [41]. Following a brief, intentional interruption of approximately 1 min for the collection of their name, sex, and age, participants proceeded to complete a series of 3 subtests. In the first part, the spatial memory subtest, participants were presented with 30 drawings of the objects in succession, with each object displayed on a screen for a duration of 4 s. During these 4 s, participants were required to use a computer mouse to indicate the object’s original position on a blank 6 × 5 grid that was also displayed on the screen. There were no pauses between the presentations. The second part, lasting 3 min, involved the free recall memory subtest, where participants were prompted to write down as many of the shown objects as they could remember. The final part, the recognition memory subtest, presented each participant with 60 object names (30 original and 30 new names), with each being presented for 4 s. They had to decide whether the picture of each object had been shown previously on the grid.

#### 2.2.3. Reaction Times

In the simple reaction-time test, participants were required to press the left button of a computer mouse in response to 40 simple visual go-stimuli. These stimuli appeared at irregular intervals of 2–8 s in the center of the computer screen. We excluded outlier values from the analysis, specifically, reaction times of longer than 700 milliseconds or shorter than 150 milliseconds [7]. However, it was noted that the results of all analyses were nearly identical even when this exclusion step was omitted.

The prepulse tests were designed similarly to the simple reaction-time test, with the addition of random, low-intensity pulses occurring 20 milliseconds before half of the go-stimuli, accompanied by either acoustic or visual white noise. In the acoustic prepulse test, the stimuli, prepulses, and white noise were all acoustic. Conversely, in the visual prepulse test, these elements were visual. The visual white noise was represented by squares of varying shades of gray that randomly appeared and expanded on the screen. The visual prepulse test also included both prepulses in the screen’s central area and occasional ‘distractors’—distinctive but brief signals—at various screen locations. In this study, we only used the data on reaction times to stimuli with and without prepulses. The investigation of the relationship between toxoplasmosis and prepulse inhibition/stimulation was addressed in a separate study [42].

#### 2.2.4. Information Processing Speed

To evaluate the participants’ information processing speed and action control, we utilized two variants of the Stroop test. In both tests, participants were instructed to react swiftly to go-stimuli and to refrain from reacting to stop-stimuli. The stimuli were the words “black”, “green”, “red”, and “yellow”, displayed in various combinations of font color and word content on the computer screen. In the A variant of the Stroop test, participants were required to respond quickly to stimuli written in black, green, and red, but not to those written in yellow. This version aimed to assess basic reaction time and inhibitory control. Conversely, the B variant was designed to test cognitive flexibility and the ability to adapt to changing rules: participants were to react to the same words in any font color except when the word was “yellow”, challenging their capacity to shift cognitive sets in response to altered task demands. Each variant included 100 stimuli. For all participants, we calculated the number of correct nonreactions (indicating action control), the number of incorrect nonreactions, and the mean reaction time of the correct responses.

### 2.3. Immunological Tests for Toxoplasma, CMV, and the Rh Phenotype

*Toxoplasma* infection status was assessed at the National Reference Laboratory for Toxoplasmosis of the National Institute of Public Health in Prague. We employed the complement fixation test (CFT, TestLine Clinical Diagnostics) and ELISA IgG test (TestLine Clinical Diagnostics) for this purpose. The ELISA assay cut-point values were established according to the manufacturer’s instructions. Subjects were considered non-infected with *Toxoplasma* if the results from both tests were negative, while those with positive results in both tests were considered *Toxoplasma*-positive. Approximately 5.3% of the participants, who did not meet the condition of having either both negative or both positive results from 2 tests, were excluded from the study before any data analysis was performed.

CMV status (infected vs. non-infected) was determined at the National Reference Laboratory for Herpes Viruses of the National Institute of Public Health in Prague, using the sandwich ELISA method with an inactivated CMV AD 169 strain antigen (ETI CYTOK G PLUS, DiaSorin, Saluggia, Italy), allowing the quantitative detection of CMV IgG from 0.4 IU/mL to 10 IU/mL. We used the threshold recommended by the manufacturer, however, we excluded about 2.8% of subjects with ambiguous serological test results (an IgG concentration between 0.36 and 0.44 IU/mL) from the study.

The Rh phenotype, specifically the presence of the RhD antigen on erythrocyte membranes, was ascertained using a commercial agglutination test with human monoclonal anti-D reagents (Seraclone1, ImmucorGamma Inc., Houston, TX, USA) during the blood collection session.

### 2.4. Data Analysis

A MANCOVA test was performed with the Statistica software, v.14. For the log-linear analysis of the 2 × 2 × 2 contingency tables (e.g., sex × infection × Rh table), we utilized an online calculator that is available at http://vassarstats.net/abc.html (accessed on 1 March 2024). All other statistical analyses, including path analysis, were conducted using the R software, version 3.3.1 [43], employing the following packages: corpcor, pcaPP, tcltk2, ppcor, psych, lavaan, and semplot. Partial Kendall correlation tests were computed in R using the Explorer 0.1 program [44].

First, we applied a threshold-based method to convert the results of the diagnostic tests (continuous variables) into binary variables—“*Toxoplasma*-infected” and “CMV-infected”. These binary variables were then utilized to assess the differences in cognitive performance between infected and non-infected subjects and in the path analyses. In the correlation analyses, we utilized continuous variables (the concentration of anti-*Toxoplasma* or anti-CMV antibodies, as measured via an ELISA) to ascertain the correlation between the concentration of antibodies and the outcomes of the cognitive performance tests. However, these analyses were conducted exclusively for the subpopulations of subjects infected with either *Toxoplasma* or CMV (independently), to prevent conflating the data from infected subjects carrying specific antibody concentrations with data from non-infected subjects carrying high levels of cross-reacting antibodies.

For an overall MANCOVA test, we employed a Type VI sum of squares (unique SS) to determine the individual contribution of each independent variable, controlling for associations with other variables in the model. This approach enabled us to isolate and assess the specific impact of each factor independently. Several variables related to cognitive performance displayed bimodal or multimodal distributions and varied between men and women, as well as between Rh-negative and Rh-positive individuals. They also showed strong correlations with participant age. Consequently, we employed nonparametric partial Kendall tests, controlling for age and sex, to investigate the associations with *Toxoplasma* and CMV infections, along with the concentrations of specific antibodies in seropositive subjects (as a proxy for the time since infection) on intelligence, knowledge, memory, reaction times, and action control. Previous studies have documented the sex-dependent associations of *Toxoplasma* infection with various behavioral variables, many of which are also dependent on the subjects’ Rh phenotype. Accordingly, our analyses were performed for the entire sample and were conducted separately for men and women and for Rh-negative and Rh-positive individuals. We controlled for the effect of multiple tests using the Benjamini–Hochberg procedure (FDR = 0.10) [45]. The complete dataset is available at Figshare: https://doi.org/10.6084/m9.figshare.21865677.

### 2.5. Technical Notes

Throughout the manuscript, we use the terms “toxoplasmosis” and “CMV infection” as shorthand for samples testing positive for the presence of anti-*Toxoplasma* or anti-CMV IgG antibodies in standard serological tests.

Given the exploratory nature of the main part of this study, we report both the results corrected and non-corrected for multiple tests. For the same reason, in the exploratory section of this study, we discuss not only the statistically significant associations but also trends that did not reach formal significance.

## 3. Results

### 3.1. Descriptive Statistics of the Population under Study

Our final dataset included 352 women (mean age 22.8; SD 3.6) and 205 men (mean age 24.2; SD 6.3). Although the age difference between women and men was small (Cohen’s d = 0.30), it was statistically significant (*p* = 0.004). In terms of infection prevalence, 51.8% of women and 51.5% of men were CMV-positive, while toxoplasmosis prevalence was 17.4% in women and 17.6% in men. Fisher’s exact test revealed that the associations between sex and toxoplasmosis, sex and CMV, and toxoplasmosis and CMV were not statistically significant (*p*-values: 0.99, 0.93, and 0.57, respectively). In all groups, *Toxoplasma*- and CMV-infected subjects were generally older than their noninfected counterparts. However, this age difference was significant only for CMV-infected men (23.1 vs. 25.1, Cohen’s d = 0.32, and *p* = 0.028).

The proportion of Rh-negative individuals was 19.3% in women and 19.1% in men, indicating that there was no significant sex difference. In Rh-positive subjects, CMV infection rates were 51.7% for women and 55.4% for men, whereas in Rh-negative individuals, the rates were 52.3% for women and 33.3% for men. Following this observation, a log-linear analysis of the 2 × 2 × 2 contingency table (sex × infection × Rh) indicated that the interaction between the Rh factor, CMV infection, and sex was not statistically significant (G^2^ = 5.8; degrees of freedom = 2, *p* = 0.055). Given the number of tests conducted, the observed difference in CMV prevalence among Rh-negative men and women is likely to be due to chance. No similar trend was observed with *Toxoplasma* (G^2^ = 3.16; degrees of freedom = 2, *p* = 0.206).

Descriptive statistics for variables related to cognitive performance, as well as the correlations of these variables with the subjects’ ages, are detailed in Table 1. This table also includes the number of students who participated in each specific performance test.

### 3.2. Associations among Toxoplasmosis, CMV Infection, Rh Phenotype, and Sex, and Their Interactions with Cognitive Performance

Before proceeding with more detailed analyses, it was necessary to determine whether *Toxoplasma* and CMV infections have a discernible impact on cognitive performance and if this impact is modulated by the Rh phenotype and sex. Therefore, we began our analysis with an overall MANCOVA test. The dependent variables were the results of all performance tests (see Table 1), with toxoplasmosis, CMV infection, Rh phenotype, and sex as binary independent variables, and age as a continuous covariate. The results are presented in Table 2. For toxoplasmosis, only the association with toxoplasmosis was significant (*p* = 0.040, partial eta² = 0.175). For CMV infection, the association with CMV infection was of similar strength but was non-significant (*p* = 0.071, partial eta² = 0.164); however, significant associations were observed for the double interactions of sex–CMV (*p* = 0.002, partial eta² = 0.223) and CMV–Rh (*p* = 0.022, partial eta² = 0.185), as well as the triple interaction of sex–CMV–Rh (*p* = 0.002, partial eta² = 0.223). The direction of these associations for each variable will be elucidated in the exploratory section of our study.

### 3.3. The Association of Toxoplasma and CMV Infections with the Different Facets of Cognitive Performance

We examined the association between the cognitive performance test outcomes and *Toxoplasma* and CMV infections using partial Kendall correlation tests, adjusting for age and sex variables. Table 3 presents the strength of associations (a partial Kendall’s Tau) and their significance, both before and after adjustments for multiple tests. Our results reveal that toxoplasmosis significantly impacts various facets of intelligence in men, specifically numerical, figural, fluid, and general intelligence, as well as several psychomotor performance metrics. In contrast, CMV infection did not demonstrate a notable association with intelligence measures. However, it notably influenced memory, as determined by three distinct memory tests, with significant findings being recorded particularly among women. Additionally, CMV infection was found to affect reaction times and action control in a way similar to *Toxoplasma* infection, although these associations were more pronounced in women compared to men.

### 3.4. Correlations between the Concentrations of Specific Anti-Toxoplasma and Anti-CMV Antibodies and Cognitive Performance

Table 3 also shows the associations of anti-*Toxoplasma* IgG and anti-CMV IgG antibody concentrations with the cognitive performance of infected subjects. Within the context of toxoplasmosis, specific IgG antibody levels may act as a proxy for the time that has elapsed since infection [46], a hypothesis we also tested for CMV. In *Toxoplasma*-infected men, high concentrations of anti-*Toxoplasma* antibodies, which are characteristic of the disease’s earlier stages, were associated with reduced intelligence, impaired memory, and faster reaction times in most tests, suggesting that as the infection progresses beyond its acute phase, the negative associations with intelligence and memory tend to decrease, whereas the associations with psychomotor performance, as evidenced by prolonged reaction times, tend to increase. However, these associations, though relatively strong, did not achieve statistical significance, likely due to the limited number of infected male subjects in our study. In women, a similar trend was observed, but the observed associations were generally milder. An exception was found in the acoustic prepulse test, where a stronger and significantly negative correlation between antibody concentration and reaction time was noted in women compared to men. The most notable sex difference emerged in memory test performance: men exhibited improved performance with decreasing anti-*Toxoplasma* IgG antibodies, presumably as the amount of time since infection increased, whereas women showed decreased performance with lower antibody levels.

The associations between anti-CMV antibodies and cognitive performance were considerably weaker and did not display a consistent pattern for either sex. However, when considering the modifying impact of the Rh phenotype, a more distinct pattern began to emerge, as will be discussed in subsequent sections.

### 3.5. The Mechanism of Improved Action Control in Toxoplasma- or CMV-Infected Individuals

Previous studies on toxoplasmosis have indicated that *Toxoplasma*-infected subjects exhibit superior action control, which is defined as the ability to inhibit a response to a false stimulus, compared to noninfected individuals [47]. In line with this finding, our data revealed that men infected with *Toxoplasma* and women infected with CMV displayed significantly longer reaction times to go-stimuli but committed fewer errors in responding to stop-stimuli, as shown in Table 2. To explore whether prolonged reaction times could be a proximate cause of the reduced error rate in infected subjects, we employed structural equation modeling, specifically, path analysis (PA). The PA results, which are illustrated in Figure 2, indeed showed low (and negative) path coefficients linking infection status to the number of correct non-reactions, along with high, significantly positive path coefficients connecting reaction times to correct non-reactions.

### 3.6. Influences of the Rh Phenotype

To further explore these connections, we conducted partial Kendall correlation tests that were controlled for age separately for Rh-positive and Rh-negative men and women. The outcomes of these analyses regarding *Toxoplasma* and CMV infections are presented in Table 4. It is important to note that the number of Rh-negative subjects in our study, which mirrored the natural occurrence of Rh-negative individuals in the population, was four times lower than that of Rh-positive subjects. As a result, many of the associations in the Rh-negative subgroup did not achieve formal statistical significance. However, the strength of the associations in Rh-negative individuals was generally, albeit not consistently, larger than those in Rh-positive individuals. Notably, the patterns of these associations varied, depending on the type of infection (*Toxoplasma* or CMV), the sex of the subjects, and across different cognitive performance tests.

## 4. Discussion

### 4.1. Confirmatory Part of the Study

The primary aim of our study was to compare the associations of two neurotrophic pathogens—*Toxoplasma*, a parasite known to manipulate its host’s behavior, and human CMV, a virus not associated with behavioral manipulation—with various cognitive functions. We aimed to evaluate their influence on intelligence, memory, reaction times, and information processing speed, as measured through a series of cognitive tests, with the assumption that *Toxoplasma* would have a more pronounced or evident association. Contrary to our expectations that significant cognitive impairments would be associated solely with *Toxoplasma*, our findings revealed that both *Toxoplasma* and CMV were associated with predominantly adverse effects and symptoms of similar magnitude on the cognitive performance of subjects. The strength of the associations of *Toxoplasma* and human CMV with various cognitive functions varied, depending on the pathogen involved, and was influenced by the sex and Rh factor status of the individuals. Specifically, the association with toxoplasmosis was more pronounced and widespread in men, while the association with CMV infection was stronger and more prevalent in women. In line with previous research [32,33,34,35], the associations were typically stronger in Rh-negative individuals and were either weaker or non-existent in Rh-positive individuals.

The formation of lifelong dormant stages that are capable of reactivation in the brain during immunosuppression or following a head injury is a characteristic shared by both *Toxoplasma* and CMV [48,49,50,51]. However, the modes of transmission of *Toxoplasma* and CMV differ. While *Toxoplasma* is transmitted through predation [52], a process that could potentially be facilitated by the impairment of a host’s cognitive abilities, CMV is transmitted through direct contact, such as with saliva, urine, milk, and genital secretions [53], and is not presumed to manipulate host behavior to enhance its transmission. Given that we observed lower cognitive performance in individuals infected with *Toxoplasma*, as well as those with CMV, it seems plausible that the differences noted between infected and non-infected subjects are more likely a result of pathological processes occurring locally in the brain due to the presence of the dormant stages of both pathogens [36,37], rather than as the result of their manipulative activity aimed at facilitating transmission from infected to uninfected hosts. However, it is important to acknowledge that this suggestion does not constitute definitive proof but, rather, indicates a possible direction for further investigation.

### 4.2. Exploratory Part of the Study

#### 4.2.1. Intelligence and Toxoplasmosis

Our study indicated that toxoplasmosis negatively influenced several aspects of intelligence, encompassing general, numerical, figural, and fluid intelligence. Notably, fluid intelligence, which is associated with problem-solving abilities, comprehension, and reasoning, was significantly affected. In contrast, its counterpart, crystallized intelligence—which relies on the recall of stored knowledge and past experiences—did not exhibit notable differences between *Toxoplasma*-infected and non-infected individuals. The adverse symptoms of toxoplasmosis on intelligence were primarily observed in men, while women exhibited either weaker or no such symptoms. Intriguingly, the Rh factor played only a minor role in the association between *Toxoplasma* infection and intelligence, unlike the impact that was observed in other performance tests or with CMV infection. The associations seen in Rh-negative and Rh-positive subjects largely differed in magnitude rather than in direction. It is important to mention that the strength of the associations, as indicated by Tau values, was typically larger in Rh-negative subjects; however, the sample size in this group was four times smaller, which inevitably limited the power of the tests and, thus, our chance to detect significant results.

Our study did not find a formally significant association of anti-*Toxoplasma* IgG concentration with the intelligence of *Toxoplasma*-infected students, yet all the observed trends were negative, indicating a negative association between the levels of anti-*Toxoplasma* IgG concentration and intelligence. This suggests that intelligence levels may be lowest shortly after infection, coinciding with the highest concentration of anti-*Toxoplasma* antibodies, and may potentially improve over time post-infection. This pattern aligns with previous findings showing a negative correlation between anti-*Toxoplasma* antibody concentration and intelligence, which was significant in that study [54,55]. Such a correlation suggests that the observed decrease in intelligence among *Toxoplasma*-positive men could be temporary, stemming from the acute phase of the infection, rather than being due to the cumulative effects of latent toxoplasmosis. However, an alternate explanation for the negative trends is that the antibody concentration may reflect not only the time elapsed since contracting toxoplasmosis but also the intensity of the infection, which could correlate with the extent of brain damage.

The impact of toxoplasmosis on intelligence remains a complex subject. While certain studies have suggested a negative influence on performance in IQ tests, others have reported contrasting results [54,55]. This variation in findings could be attributed to the wide range of both direct and indirect associations that *Toxoplasma* infection has with human behavior and physiology. Beyond its impact on intelligence, the infection has been linked to psychological traits like increased competitiveness, possibly due to increased testosterone levels in men [56], and altered levels of cooperativeness—decreasing in men while increasing in women [57]. Both competitiveness and cooperativeness can significantly affect outcomes on IQ tests. For certain subpopulations and with specific aspects of intelligence, such as verbal intelligence, the positive influences of these traits might counterbalance the negative associations between the infection and cognitive performance, including performance in IQ tests.

#### 4.2.2. Intelligence and CMV Infection

The association between CMV infection and cognitive performance, particularly in young and middle-aged individuals, has not been studied extensively. The present study found many adverse symptoms of CMV infection related to various facets of intelligence. They were particularly pronounced in Rh-negative men; however, due to the low representation of Rh-negative individuals in the sample (19%) and the underrepresentation of men among the university students (36.7%), the size of this specific group was relatively small (24 CMV-free and 12 CMV-infected individuals). Consequently, most of the observed associations remained formally nonsignificant in the Rh-negative subpopulation, with the exception of lower general and figural intelligence. In this context, it is worth noting that the partial Kendall correlation is an “exact test”, a term referring not only to its precision but also to a category of statistical analyses that are robust, even with limited or unevenly distributed data.

In contrast to the findings related to *Toxoplasma*, the analysis did not reveal any consistent pattern in the correlations between anti-CMV IgG antibody concentration and the intelligence of CMV-infected subjects. We observed an equal number of positive and negative trends. Such a pattern might be expected if the decline in intelligence post-infection was relatively rapid and concluded within a few years or even months after CMV infection. However, it is crucial to remember that unlike with toxoplasmosis [46], no study to date has shown a correlation between anti-CMV IgG antibody concentration and the time since infection. In fact, some evidence suggests that anti-CMV IgG levels may increase over time after the initial CMV infection, or at least increase with the age of the patient [58,59].

#### 4.2.3. Toxoplasmosis and Memory

Research into the association between toxoplasmosis and memory functions in young and healthy individuals has been relatively scarce, compared to its association with intelligence. A recent meta-analysis [60] reviewed six articles addressing the impact of toxoplasmosis on working memory, revealing an aggregate effect size (standardized mean difference, SMD) of 0.16 with a *p*-value of 0.002. Additionally, it identified five articles that studied short-term word memory, showing an aggregate effect of 0.18 with a *p*-value of less than 0.001. However, except for one study [61], these analyses predominantly focused on senior or middle-aged populations.

In our current study, we assessed the memory performance of *Toxoplasma*-infected and *Toxoplasma*-free university students using four different memory tests. Our findings indicated mostly negative, yet nonsignificant, associations of toxoplasmosis with the students’ memory performance. The lack of a significant overall association with memory might be attributed to toxoplasmosis having opposite impacts on Rh-positive and Rh-negative subjects. Notably, this association was more pronounced in male students: Rh-positive-infected males tended to perform worse, while Rh-negative-infected males performed better than their *Toxoplasma*-free male counterparts in three out of the four memory tests. Again, due to the lower prevalence of Rh-negative subjects in our study sample, these observed associations were significant only in the I-S-T 2000 R memory test and not in any variant of the Meili tests. Similar patterns were observed in female students, although the associations were weaker and did not reach statistical significance.

#### 4.2.4. CMV and Memory

Research on the impact of CMV infection on memory functions has predominantly focused on patients with dementia or on seniors [14,59,62,63,64,65,66]; however, some studies have deviated from this trend [16,65]. For instance, a longitudinal study over 5 years, involving 1022 older individuals from the Monongahela–Youghiogheny Healthy Aging Team, assessed cognitive decline annually using various cognitive tests, including memory tests. This study revealed that CMV-infected individuals experienced a significantly greater cognitive decline over time compared to their non-infected counterparts [21]. Conversely, Gale et al. [66] reported no association between CMV and memory in a community-based sample of adults aged 40 to 70 years.

In our current study, we found that CMV-infected female students, unlike their male counterparts, performed significantly worse than non-infected peers in three of the four memory tests. Further analysis, stratified by the Rh factor, showed that this pattern was much more pronounced in Rh-negative women.

#### 4.2.5. Toxoplasmosis and Reaction Times

The first study reporting the associations of toxoplasmosis with reaction times was published in 2001 [10]. It showed that *Toxoplasma*-infected blood donors had longer reaction times than *Toxoplasma*-free blood donors and that in *Toxoplasma*-infected individuals, the reaction times correlated negatively with concentrations of anti-*Toxoplasma* IgG antibodies (which decreased with the amount of time passed since infection). This suggests that reaction times worsen as more time elapses following the acute phase of toxoplasmosis. This could indicate that rather than being the aftereffects of acute toxoplasmosis, we are seeing a cumulative effect of latent toxoplasmosis.

Later studies demonstrated that the associations of toxoplasmosis with the reaction times of blood donors and students depend on the subjects’ Rh factor status [32,35]. Rh-negative *Toxoplasma*-free subjects (Toxo−Rh-) had the shortest reaction times, while Rh-negative *Toxoplasma*-infected subjects (Toxo+Rh-) had the longest reaction times from all four groups (Toxo+Rh+, Toxo+Rh-, Toxo−Rh+, Toxo−Rh-) of subjects. Moreover, the results showed that the association depended on the Rh genotype rather than on the Rh phenotype—the mitigating relationship of Rh-positivity with the behavioral symptoms of toxoplasmosis was more evident in Rh-positive heterozygotes than in Rh-positive homozygotes [29]. These findings, among others, have prompted the theory that Rh polymorphism may be maintained in human populations through the selective advantages conferred upon heterozygotes in environments where *Toxoplasma* infection is common, and where, until recently, it was nearly ubiquitous [29,30,35].

In our current study, we found no substantial evidence to suggest any significant association between toxoplasmosis and reaction times as measured by the simple reaction-time test. However, the prepulse tests revealed predominantly negative correlations (or trends) between the concentration of anti-*Toxoplasma* IgG antibodies and reaction times. This pattern implies that reactions tend to become slower as more time passes since the infection. Such findings suggest that the cumulative effects of latent toxoplasmosis, rather than the diminishing effects of past acute toxoplasmosis, might be contributing to the observed changes in reaction times.

Additionally, the Stroop tests revealed a significant positive association between toxoplasmosis and reaction times, indicating an impairment of reaction times correlating with the length of time since infection. This association was more pronounced in men, with variations being observed between Rh-positive and Rh-negative individuals. However, this result was not consistent across the two versions of the Stroop test, introducing an element of uncertainty into its interpretation. Similar to the prepulse tests, the Stroop tests also a exhibited negative correlation (or strong, though nonsignificant, trends) of the concentrations of anti-*Toxoplasma* IgG antibodies with reaction times.

#### 4.2.6. Toxoplasmosis and Action Control

An essential outcome from the Stroop tests in our study was the measurement of the number of “correct nonreactions”. Notably, *Toxoplasma*-infected men, but not women, exhibited better scores in this metric compared to the *Toxoplasma*-free controls. Further analysis, stratified according to Rh factor, indicated that Rh-negative individuals, particularly men, were primarily responsible for this association. A higher number of correct nonreactions is typically interpreted as being indicative of better action control, meaning the subject’s ability to halt an already initiated reaction to a false stimulus.

Previous studies have reported and discussed the enhanced performance of *Toxoplasma*-infected subjects in tasks requiring strong action control [47,67]. However, the results of our path analyses suggest that this improved performance in the test might be a secondary outcome of their longer (worse) reaction times. In essence, the infected subjects do not necessarily exhibit better action control; rather, they have more time to adjust their reaction due to their slower responses to stimuli.

#### 4.2.7. CMV and Reaction Times

CMV-infected subjects, particularly women, exhibited worse performance on nearly all reaction-time tests, with the exception of the acoustic prepulse tests. This pattern was more pronounced in Rh-negative individuals compared to Rh-positive ones. Specifically, in the first Stroop test (version A), CMV-infected women demonstrated a higher number of correct non-reactions. However, among men, those who were Rh-positive and CMV-infected scored lower for action control compared to their Rh-positive, CMV-free counterparts.

From a broader perspective, the observed impact of CMV infection on reaction times and cognitive performance introduces significant skepticism toward the hypothesis that exclusively associates the existence of Rh polymorphism in Europe with historical fluctuations in toxoplasmosis prevalence [32,35]. Our findings, by demonstrating comparable Rh-dependent interactions in the context of CMV infection, suggest that attributing Rh polymorphism solely to shifts in toxoplasmosis prevalence may be an oversimplification or may potentially be incorrect.

### 4.3. Strength and Limitations

This study exhibits several significant strengths, as well as certain limitations. A primary strength, particularly in comparison with similar studies, as underscored in a recent meta-analysis [60], is the considerable number of participants involved. This larger sample size significantly enhanced the statistical power of our analyses. However, the stratified analyses in the explorative part of the study encountered challenges, especially due to the small sizes of certain subgroups, such as Rh-negative *Toxoplasma*-infected men, as outlined in Table 3. As a result, the findings from some Rh-stratified analyses should be approached with caution. Additionally, the limited representation of specific participant classes in our sample restricted our ability to include more factors in our analytical models. For example, the current sample size did not permit a thorough investigation into the impacts of CMV–*Toxoplasma* and CMV–*Toxoplasma*–Rh interactions on cognitive performance. This underscores the need for future studies with larger and more diverse cohorts to delve deeper into these intricate interactions and their impact on cognition.

Previous research on the relationship between latent toxoplasmosis and intelligence often relied on relatively simple IQ tests, focusing primarily on facets like verbal or figural intelligence [54,55]. The key advantage of our current study is the use of the comprehensive I-S-T 2000 R questionnaire, a four-hour-long psychometric tool that measures or calculates a wide range of intelligence facets, as well as knowledge and memory.

A particular consideration in our study was the time gap between performance testing and the serological testing for pathogens in some students. While this is less of a concern for *Toxoplasma*, due to the relatively low incidence of infection in this age group, it poses a potential issue for CMV. The incidence of CMV infections is relatively high in students, and it is conceivable that an unknown number of them might have acquired a CMV infection during the period between taking the serological and performance tests. The inclusion of such CMV-positive individuals with initially negative serological test results could have attenuated the observed differences between CMV-infected and CMV-free subjects. Nevertheless, a numerical simulation indicated that while this scenario might increase the risk of a Type II error (failing to identify weak associations), it would not elevate the risk of a Type I error (identifying non-existent associations) [68].

Similarly, our dataset must contain some false negative individuals, i.e., individuals who tested negative in the serological test but who were, in fact, infected. The serological tests that we used are considered reliable and are utilized not only for research purposes but also in clinical practice. The reference laboratories at the National Health Institute represent top diagnostic facilities for toxoplasmosis and herpesviruses, setting (and overseeing) the standard for other diagnostic facilities in the Czech Republic. Even so, it is certain that at least some of the individuals who tested negative are, in fact, infected; due to the long interim between infection and testing, their antibody levels could have fallen below the threshold value. Again, such an error could cause a Type II error, but it would not elevate the risk of a more serious Type I error [68].

The fundamental experimental design of our study was cross-sectional, presenting challenges in distinguishing between causes and effects. However, our analysis went beyond merely assessing the impact of seropositivity for *Toxoplasma* or CMV. We also explored the correlation between the concentrations of specific anti-*Toxoplasma* and anti-CMV IgG antibodies. In the case of toxoplasmosis, these antibody concentrations serve as a statistical proxy for the time that has elapsed since infection, as demonstrated in an earlier paper [46]. Such correlations, combined with laboratory infection results in rodents for *Toxoplasma*, strongly suggest that the infection is likely to be the cause, rather than the effect, of the observed differences in cognitive performance [69]. An alternative explanation could involve an unknown third factor, such as socioeconomic status, correlating with the likelihood of CMV or *Toxoplasma* infection and intelligence. While the Czech population, particularly students of the Faculty of Science at Charles University, exhibits a relatively low level of socioeconomic stratification, future studies should consider a broader range of potential confounding variables.

In the present study, we used the concentration of specific IgG antibodies as a proxy for the duration of infection. We would like to remind readers that this proxy is relatively tentative, as this concentration may fluctuate or remain relatively stable in many subjects over time. Generally, however, after a rapid increase during the first 1–2 months post-infection, the concentration of IgG antibodies in most individuals decreases over the duration of latent toxoplasmosis. Therefore, in research studies conducted on sufficiently large samples, the level of IgG antibodies can be used to estimate the duration of latent toxoplasmosis. It is important to emphasize that the level of these antibodies cannot be used to estimate the duration of infection in individuals, for example, in pregnant women, where entirely different methods must be used that are based on measurements over time, on antibody avidity measurements, and on monitoring the clinical symptoms.

In the current study, we did not include individuals with borderline concentrations of anti-CMV antibodies (0.36–0.44 IU/mL) or individuals whose results from CFT and ELISA tests for toxoplasmosis did not match. In the case of CMV, visual inspection of the distribution of antibody concentration values indicates that we likely only eliminated a small number of infected individuals, suggesting that this procedure had only a minimal impact on the obtained results. However, in the case of toxoplasmosis, the boundary between the antibody concentration values of negative and positive individuals was much more blurred; by filtering, we likely eliminated a much larger number of individuals infected with *Toxoplasma*. These were predominantly individuals who had been exposed to the infection for the longest period and, thus, were probably the most strongly affected by the infection. Therefore, it can be assumed that the actual strengths of the associations between latent toxoplasmosis and cognitive performance are higher in the general population than in those measured in our censored sample. On the other hand, our many years of experience show that in some uninfected individuals, especially, the levels of antibodies cross-reacting with *Toxoplasma* antigens in the CFT test are transiently or permanently elevated, which is likely due to ongoing inflammatory processes. These elevated levels of cross-reacting antibodies can also affect the results of ELISA tests. Including such falsely positive individuals in the study sample would likely significantly reduce the strength of associations between infection and cognitive performance, and would additionally completely devalue the results of tests studying the correlations of antibody concentration with performance test outcomes.

Some significant associations observed in the total population (unstratified by Rh factor and sex) may seem small, yet many associations were surprisingly strong for the field of biological psychology. For instance, a Tau value of 0.161 for the association between toxoplasmosis and general intelligence in men corresponds to a Cohen’s f of 0.26, which is considered a medium magnitude [70]. Similarly, a Tau of −0.32 for the association between CMV and general intelligence in Rh-negative men corresponds to a Cohen’s f of 0.55, which is typically deemed a large magnitude.

Given the exploratory nature of the main part of this study, we discussed not only the formally significant results but also the nonsignificant trends and results that lost significance after correcting for multiple tests. Confirming these results in larger cross-sectional studies or in case-control studies enriched with Rh-negative *Toxoplasma*-infected individuals is essential.

While this sample from the Faculty of Science at Charles University, a prestigious institution, may offer insights into its student population, it is crucial to acknowledge that these students, with an average IQ of 120, might not accurately represent the broader Czech population. Consequently, extrapolating the results of our study to a wider context should be approached with caution.

## 5. Conclusions

We have substantiated that *Toxoplasma* infection is associated with an effect on various aspects of human cognitive performance, many of which are dependent on sex and Rh phenotype. Similarly, we have extended these findings to another pathogen, the human cytomegalovirus (CMV). Given the high prevalence of latent toxoplasmosis (about 33% worldwide) and CMV infection (likely over 80% in adult populations), the observed association, such as a decrease of 2.3 IQ points in *Toxoplasma*-infected individuals compared to uninfected ones (with a Cohen’s d of 0.213, categorized as a small magnitude), could have significant implications for the quality of life of a large number of people. These differences, observed as a decrease in IQ among infected individuals, may not only have implications for those who are directly affected but could also indirectly influence the broader uninfected population through societal and communal interactions.

Currently, there are no available treatments for latent toxoplasmosis or CMV infection. However, the development of vaccines against these infections could offer a preventative solution. Specifically in terms of toxoplasmosis control, vaccinating both pet and feral cats with oral vaccines could be a pivotal strategy in eradicating the disease. Our study highlights the importance of continued research into pathogens that may seem harmless but that could have significant impacts on human health and well-being.

## Figures and Tables

**Figure 1 pathogens-13-00363-f001:**
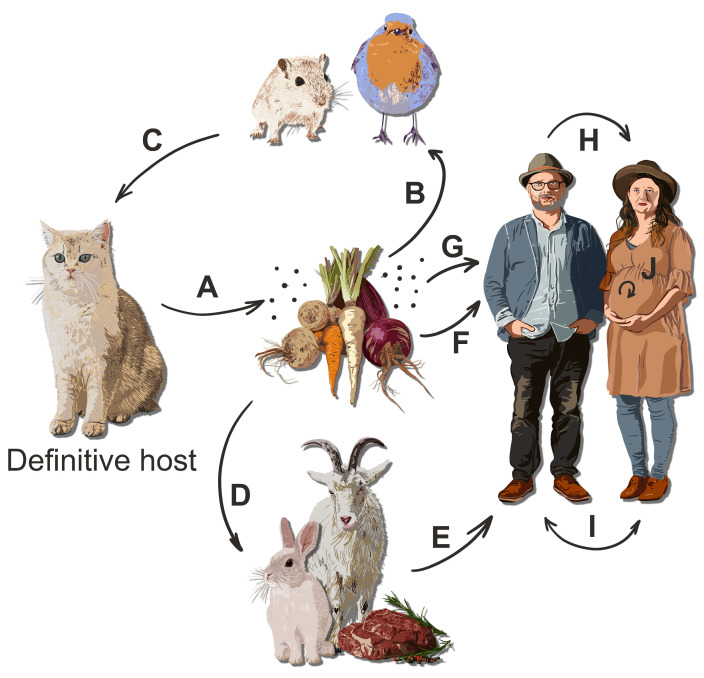
Transmission of *Toxoplasma gondii* through traditional and recently described pathways. (A) Environmental contamination with oocysts affects soil, water, plant life, and, potentially, air. (B) Small animals contract toxoplasmosis from ingesting contaminated food and environmental exposure. (C) Predation leads to the transfer of the parasite as the definitive hosts consume infected prey. (D) Livestock become infected via contaminated feed and environmental sources, serving as intermediate hosts. (E) Humans risk infection by consuming undercooked meat harboring tissue cysts and unpasteurized dairy products with tachyzoites. (F) Human infection can also occur from contaminated vegetables, water, or other foods such as fish from contaminated sources, with potential direct infection from contact with feline feces. (G) Airborne transmission might be possible for humans and animals in highly contaminated environments [2]. (H) Sexual transmission in humans, specifically through penile–vaginal [3] or, potentially, oral routes [4] can be significant due to the risk of congenital toxoplasmosis. (I) Transmission through blood transfusion or organ transplantation is an acknowledged risk, albeit tempered by the stringent screening processes in place. (J) Congenital toxoplasmosis results from maternal–fetal transmission and represents a significant route of infection, with implications for neonatal health.

**Figure 2 pathogens-13-00363-f002:**
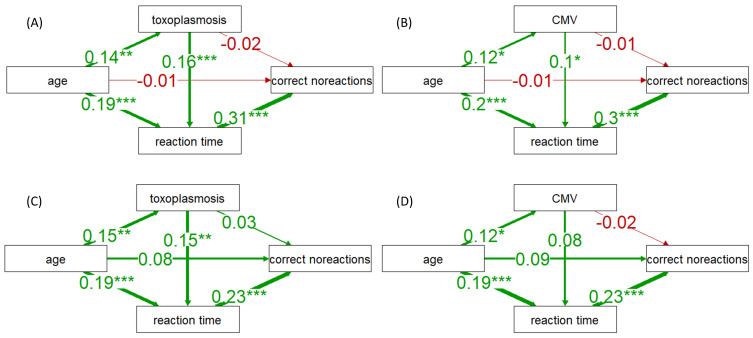
The results of path analyses showing direct and reaction time-mediated associations between *Toxoplasma* and CMV infections and the subjects’ performances in the Stroop tests. Schemes (**A**,**B**) show the results (path coefficients) for Stroop test A; schemes (**C**,**D**) show the results for Stroop test B. One, two, and three asterisks indicate associations at the levels of 0.05, 0.01, and 0.001, respectively.

**Table 1 pathogens-13-00363-t001:** Average cognitive performance metrics by sex, *Toxoplasma* and CMV infection status, and age correlations.

	Sex		*Toxoplasma*		CMV		Age		
	Women	Men	Free	Infected	Free	Infected	All	Men	Women
**I-S-T 2000 R tests (intelligence and knowledge: N = 533, memory: N = 305)**									
General intelligence	118.2	121.5	119.8	117.5	119.3	119.6	**−0.076**	−0.046	**−0.100**
General knowledge	115.9	122.8	118.6	118.1	118.6	118.3	0.040	**0.145**	−0.028
Verbal intelligence	130.1	131.7	130.8	130.3	129.9	131.4	0.036	−0.002	0.064
Verbal knowledge	111.4	117.2	113.5	114.0	113.0	114.3	**0.098**	**0.209**	0.031
Numerical intelligence	108.6	114.1	111.1	108.6	110.9	110.6	**−0.066**	−0.037	**−0.091**
Numerical knowledge	115.0	120.3	117.4	115.1	117.7	116.3	−0.052	−0.011	**−0.086**
Figural intelligence	109.9	110.6	110.6	108.0	110.3	109.9	**−0.119**	−0.080	**−0.152**
Figural knowledge	116.1	123.1	118.6	119.0	118.8	118.4	0.051	**0.131**	0.004
Crystallized intelligence	125.6	132.6	128.2	128.3	128.2	128.2	**0.060**	**0.166**	−0.006
Fluid intelligence	122.9	125.9	124.5	121.7	123.9	124.1	**−0.105**	−0.083	**−0.124**
Associative memory	17.9	16.7	17.4	17.4	17.7	17.2	**−0.141**	−0.101	**−0.166**
**Meili memory tests (location: N = 457, free recall: N = 454, yes-no: N = 456)**									
Spatial memory	14.3	12.5	13.7	13.2	13.9	13.3	**−0.067**	−0.066	−0.066
Free recall memory	13.7	13.1	13.5	13.1	13.6	13.3	**−0.098**	**−0.170**	−0.036
Recognition memory	54.8	54.7	54.6	55.5	54.8	54.7	−0.059	**−0.139**	0.006
**Simple reaction-time tests (N = 466)**									
Reaction time	279.1	271.7	275.8	279.6	274.4	278.2	−0.048	−0.026	−0.066
**Prepulse tests (acoustic: N = 415, visual: 420)**									
Acoustic test, prepulse	369.1	365.5	366.0	377.1	366.5	369.6	**0.072**	**0.129**	0.025
Acoustic test, no prepulse	401.1	398.7	398.8	408.2	400.6	400.4	**0.067**	**0.157**	−0.001
Visual test, prepulse	264.8	262.6	264.1	264.1	260.5	267.0	**0.066**	**0.142**	0.015
Visual test, no prepulse	270.0	265.8	268.4	269.2	265.8	270.7	0.025	0.062	−0.004
**Stroop tests (test A: N = 419, test B: N = 407)**									
Test A, correct no-reactions	15.9	15.4	15.6	16.2	15.6	15.9	0.059	0.039	0.068
Test A non-correct no-reactions	0.3	2.3	0.9	1.4	0.8	1.3	−0.030	−0.065	0.000
Test A reaction time	363.6	372.8	363.7	382.7	362.5	371.8	0.052	0.107	0.021
Test B correct no-reactions	6.9	7.0	6.9	7.1	6.9	7.0	0.041	**0.135**	−0.032
Test B non-correct no-reactions	0.4	0.9	0.3	1.6	0.7	0.2	−0.001	−0.033	0.020
Test B reaction time	419.5	440.9	423.9	445.2	422.6	432.6	0.042	0.065	0.028

Columns 2 to 7 present the mean scores of the various performance tests, including IQ, the number of correctly recalled stimuli in memory tests, reaction times in milliseconds, and the counts of correct and incorrect reactions. Columns 8 to 10 detail the impact of the participants’ ages on their test performance, quantified using a partial Kendall’s Tau. Taus that are significant (*p*-values < 0.05) after adjusting for multiple tests are highlighted in bold. Additionally, the number of subjects who participated in each test is indicated in the respective headers of the table’s different sections. For a detailed examination of the magnitude and potential significance of the associations observed in the univariate tests, readers are directed to Appendix A. The results from the corresponding multivariate tests, which incorporate adjustments for sex and age, are discussed in a subsequent section of the study.

**Table 2 pathogens-13-00363-t002:** Associations of various biological factors with a cognitive performance–multivariate MANCOVA test.

	F_25,190_	*p*	Partial eta^2^
Intercept	933.7954	**0.000**	0.992
sex	2.0472	**0.004**	0.212
toxo	1.6092	**0.040**	0.175
cmv	1.4916	0.071	0.164
Rh	0.8422	0.684	0.100
sex*toxo	0.8322	0.697	0.099
sex*cmv	2.1844	**0.002**	0.223
toxo*cmv	1.1808	0.261	0.134
sex*Rh	0.8172	0.717	0.097
toxo*Rh	1.2188	0.227	0.138
cmv*Rh	1.7244	**0.022**	0.185
sex*toxo*cmv	0.9333	0.559	0.109
sex*toxo*Rh	0.8077	0.729	0.096
sex*cmv*Rh	1.6449	**0.033**	0.178
toxo*cmv*Rh	0.9956	0.475	0.116
sex*toxo*cmv*Rh	1.0808	0.368	0.125
age	2.9912	**0.000**	0.282

This table presents the outcomes of the MANCOVA test (using the Type VI sum of squares), which included the results of all performance tests as dependent variables. Age was incorporated as a covariate, while toxoplasmosis, CMV infection, Rh phenotype, and sex were used as independent variables. Statistically significant associations are highlighted in bold. For clarity, significance levels (*p*-values) lower than 0.0005 are reported as 0.000 in the table.

**Table 3 pathogens-13-00363-t003:** The association of *Toxoplasma* and CMV infections with performance in cognitive tests.

	All *Toxoplasma*		Women *Toxoplasma*		Men *Toxoplasma*		All CMV		Women CMV		Men CMV	
	Infection	IgG	Infection	IgG	Infection	IgG	Infection	IgG	Infection	IgG	Infection	IgG
**I-S-T 2000 R**												
General Intelligence	**−0.071**	−0.031	−0.015	−0.021	** −0.161 **	−0.075	0.011	0.062	0.023	0.055	−0.006	0.059
General knowledge	−0.029	−0.058	0.019	−0.047	** −0.124 **	−0.069	−0.011	−0.014	−0.041	0.001	−0.016	−0.082
Verbal intelligence	−0.005	−0.003	0.004	0.036	−0.014	−0.080	0.053	0.020	0.047	0.013	0.055	0.027
Verbal knowledge	0.007	−0.037	0.029	−0.059	−0.037	−0.001	0.046	−0.018	0.017	−0.034	0.077	−0.001
Numerical intelligence	**−0.059**	−0.001	0.012	−0.017	** −0.160 **	0.002	0.004	0.054	0.013	0.073	−0.004	0.021
Numerical knowledge	**−0.062**	−0.094	−0.018	−0.112	** −0.133 **	−0.089	−0.042	−0.001	−0.071	0.023	−0.022	−0.044
Figural intelligence	**−0.058**	−0.021	−0.022	−0.005	** −0.121 **	−0.048	−0.004	0.044	0.005	0.016	−0.027	0.102
Figural knowledge	0.002	−0.007	0.042	0.089	−0.082	−0.166	−0.014	−0.032	−0.024	−0.028	−0.007	−0.056
Crystallized intelligence	−0.010	−0.054	0.020	−0.056	−0.079	−0.069	−0.002	−0.040	−0.034	−0.034	0.013	−0.090
Fluid intelligence	**−0.063**	−0.011	−0.001	0.003	** −0.165 **	−0.055	0.010	0.062	0.016	0.072	0.007	0.042
Associative memory	−0.038	0.038	−0.040	0.183	−0.050	−0.153	−0.075	0.007	**−0.099**	0.021	−0.049	−0.056
**Meili memory tests**												
Spatial memory	−0.041	−0.036	−0.061	0.036	−0.017	−0.141	−0.060	0.037	** −0.149 **	−0.011	0.086	0.108
Free recall	−0.043	−0.086	−0.072	0.043	0.003	**−0.307**	−0.061	0.015	** −0.111 **	−0.014	0.029	0.051
Recognition memory	0.061	−0.003	0.056	0.074	0.075	−0.154	−0.025	0.045	−0.070	0.000	0.055	0.083
**Simple reaction-time test**												
Reaction time	0.027	−0.067	−0.012	0.100	0.086	−0.003	**0.071**	−0.033	**0.100**	0.037	0.025	**−0.157**
**Prepulse tests**												
Acoustic test, prepulse	0.064	**−0.174**	0.064	**−0.206**	0.057	−0.182	0.040	0.035	0.042	0.107	0.028	−0.041
Acoustic test, no prepulse	0.048	−0.108	0.023	**−0.201**	0.085	−0.016	0.036	0.028	0.039	0.081	0.028	−0.030
Visual test, prepulse	−0.005	−0.068	−0.012	−0.139	0.006	0.062	** 0.090 **	0.039	**0.083**	0.070	0.101	0.020
Visual test, no prepulse	0.006	−0.026	−0.010	−0.054	0.025	0.013	**0.068**	0.014	0.061	0.045	0.081	−0.005
**Stroop tests**												
Test A, correct no-reactions	0.052	−0.001	−0.046	0.028	** 0.205 **	−0.085	0.021	0.027	** 0.154 **	0.030	** −0.177 **	0.033
Test A incorrect no-reactions	0.059	−0.079	0.058	−0.126	0.060	−0.038	0.052	0.089	0.049	0.044	0.062	0.141
Test A reaction time	** 0.129 **	−0.124	**0.091**	−0.095	** 0.207 **	−0.162	** 0.117 **	0.013	** 0.165 **	0.001	0.027	0.034
Test B correct no-reactions	0.063	−0.088	0.028	0.028	** 0.124 **	−0.245	0.001	−0.030	0.005	0.029	−0.019	−0.112
Test B incorrect no-reactions	0.028	0.023	−0.010	−0.108	0.084	0.202	0.019	**−0.099**	0.053	−0.103	−0.030	−0.077
Test B reaction time	** 0.116 **	**−0.178**	0.082	−0.153	** 0.188 **	−0.243	** 0.098 **	−0.022	** 0.141 **	0.038	0.029	−0.101

This table delineates the magnitude and direction (a partial Kendall’s Tau) of the associations between infection status (binary variable: 0 for infection-free, 1 for infected) and specific antibody concentrations (quantified via ELISA; the analysis of antibody concentrations was restricted to infected subjects) on the participants’ performance in various cognitive tests. The statistical tests were controlled for age and sex variables. Statistically significant associations are highlighted in bold. Additionally, those associations that retain their significance after correction for multiple tests are marked with an underline.

**Table 4 pathogens-13-00363-t004:** The association of *Toxoplasma* and CMV infections with performance in cognitive tests in Rh-positive (Rh^+^) and Rh-negative (Rh^−^) subjects.

	Toxoplasmosis						CMV Infection					
	All		Women		Men		All		Women		Men	
	Rh^−^	Rh^+^	Rh^−^	Rh^+^	Rh^−^	Rh^+^	Rh^−^	Rh^+^	Rh^−^	Rh^+^	Rh^−^	Rh^+^
Number of subjects	107	449	68	283	39	166	101	426	65	269	36	157
General intelligence	−0.106	−0.063	−0.135	0.013	−0.089	** −0.174 **	−0.102	0.030	0.016	0.028	** −0.321 **	0.037
General knowledge	−0.097	−0.016	−0.076	0.042	−0.156	** −0.121 **	−0.072	0.011	−0.017	−0.043	−0.205	0.049
Verbal intelligence	−0.056	0.005	−0.072	0.016	−0.033	−0.015	0.056	0.049	0.118	0.031	−0.088	0.069
Verbal knowledge	−0.049	0.018	−0.083	0.060	−0.010	−0.048	−0.052	**0.076**	−0.011	0.025	−0.173	**0.139**
Numerical intelligence	−0.091	−0.053	−0.083	0.035	−0.157	** −0.163 **	−0.107	0.018	−0.055	0.026	−0.222	0.005
Numerical knowledge	−0.083	−0.056	−0.021	−0.013	−0.173	** −0.121 **	−0.120	−0.022	−0.127	−0.057	−0.105	0.005
Figural intelligence	−0.046	**−0.065**	−0.090	−0.011	−0.024	** −0.146 **	−0.085	0.012	0.008	0.007	**−0.251**	0.018
Figural knowledge	−0.091	0.019	−0.115	0.078	−0.081	−0.083	0.032	−0.017	0.077	−0.043	−0.067	0.014
Crystallized intelligence	−0.055	0.000	−0.056	0.040	−0.076	−0.080	−0.043	0.017	−0.014	−0.033	−0.086	0.059
Fluid intelligence	−0.060	**−0.065**	−0.071	0.011	−0.067	** −0.178 **	−0.085	0.022	−0.008	0.026	−0.227	0.024
Associative memory	0.125	**−0.085**	0.054	−0.058	**0.319**	** −0.148 **	−0.023	**−0.100**	−0.106	−0.101	0.067	−0.099
Spatial memory	0.028	−0.062	−0.045	−0.070	0.147	−0.061	** −0.231 **	−0.029	** −0.293 **	** −0.110 **	−0.074	0.103
Free recall memory	0.021	−0.059	−0.045	−0.080	0.154	−0.037	−0.139	−0.047	−0.189	**−0.096**	−0.066	0.045
Recognition memory	0.057	0.058	−0.005	0.066	0.146	0.051	0.048	−0.050	0.010	−0.085	0.100	0.023
Simple reaction time	−0.039	0.046	−0.070	0.004	−0.021	0.106	** 0.177 **	0.051	0.184	0.080	0.148	0.011
Reaction time acoustic test, prepulse	0.124	0.044	0.143	0.038	0.094	0.050	0.089	0.027	0.142	0.021	−0.007	0.021
Reaction time acoustic test, no prepulse	0.048	0.042	0.097	−0.002	−0.051	0.109	**0.155**	0.012	0.159	0.010	0.189	0.000
Reaction time visual test, prepulse	−0.040	0.006	−0.075	0.004	0.000	0.011	** 0.286 **	0.050	**0.253**	0.041	** 0.359 **	0.065
Reaction time visual test, no prepulse	−0.079	0.030	−0.126	0.024	−0.002	0.033	** 0.229 **	0.048	0.138	0.045	** 0.412 **	0.058
Stroop test A correct no-reactions	−0.064	**0.087**	−0.113	−0.019	0.039	** 0.234 **	0.084	0.019	0.113	** 0.174 **	0.020	** −0.203 **
Stroop test A non-correct no-reactions	** 0.267 **	0.003	**0.215**	0.022	**0.358**	−0.020	0.008	0.071	−0.095	0.081	0.102	0.072
Stroop test A reaction time	0.130	** 0.127 **	0.174	0.060	0.071	** 0.249 **	** 0.209 **	**0.102**	**0.203**	** 0.159 **	0.221	−0.009
Stroop test B correct no-reactions	0.123	0.050	0.110	0.016	0.165	0.103	0.006	−0.002	−0.050	0.026	0.104	−0.064
Stroop test B non-correct no-reactions	−0.014	0.033	−0.049	−0.010	0.078	0.088	0.001	0.029	0.066	0.035	−0.113	0.012
Stroop test B reaction time	0.031	** 0.135 **	0.033	**0.092**	0.034	** 0.222 **	** 0.188 **	**0.085**	**0.207**	** 0.133 **	0.139	0.013

The table shows the size and direction of the associations of *Toxoplasma* and CMV infections (binary variable 0: infection-free, 1: infected) with the subjects’ results in the cognitive performance tests. Significant associations (a partial Kendall’s Tau, controlled for age) are in bold; those that retained significance after correction for multiple tests are underlined.

## Data Availability

The complete dataset is available at Figshare: https://doi.org/10.6084/m9.figshare.21865677.

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
