# Peer review of "Cognitive Effects of Toxoplasma and CMV Infections: A Cross-Sectional Study of 557 Young Adults Considering Modulation by Sex and Rh Factor"

_pathogens, 2024, doi:10.3390/pathogens13050363_

Round 1
Reviewer 1 Report
Comments and Suggestions for Authors
Jaroslav Flegr and colleagues submit for publication a study entitled “Cognitive Effects of Toxoplasma and CMV Infections: A Cross- 2 Sectional Study of 557 Young Adults Considering Modulation 3 by Sex and Rh Factor”.
It investigates an interesting aspect from an innovative angle by evaluating the associations between a selection of outcomes and two types of exposure, but calls for two major comments
First, throughout the paper, reference is made to the "effects" in humans of toxoplasmosis or CMV infection. However, cross-sectional studies such as the current one, or those referred to in this document, are based on the concomitant evaluation of outcomes and exposures, with no follow-up, and do not allow conclusions to be drawn in terms of causality. This is a major point in terms of methodological and scientific rigor. This point is discussed in the discussion, and should concordantly be maintained throughout the manuscript. The numerous uses of the word "effect" should be replaced by "association".
The second point concerns the assessment of exposure to toxoplasmosis or CMV infection, which is a major issue.
Concerning toxoplasmosis
can the authors specify the cut off points for considering serological tests to be positive?
If they are those recommended by the manufacturer, this should be indicated. If they have been modified, this should also be indicated and justified.
How many patients were in doubtful zones for toxoplasmosis? How were they considered?
The authors mention that participants with discordant results between the two tests for toxoplasmosis were excluded. How many participants were involved? At what point were they excluded?
It would not have been illogical to consider these patients as positive, in the context of a long standing infection profile and of IgG levels that had regressed and fluctuated around a threshold. To what extent would considering them as positive have altered the conclusions? This point needs to be discussed.
How was antibody concentration defined, was it taken into account in a linear fashion, or were thresholds defined?
Concerning CMV,
It should be clarified whether the cut-off values used were those of the manufacturer, or if not, how they were defined and why.
Why were patients with low values excluded? It would have been interesting to analyze the data by considering them as weakly positive, to see the impact on the results.
How was antibody concentration defined, was it taken into account in a linearly, or were thresholds defined?
The authors should also comment in the discussion on the difficulty of estimating the reality of exposure to toxoplasmosis and CMV on the basis of a single sample. The entire study is based on the analysis of a single sample per patient. This is an major intrinsic limitation of this type of survey, which must be discussed, particularly as regards the lack of sensitivity of the tests used.
In addition
The introduction could be shortened, and certain elements included in the discussion.
The results section contains many elements that should be placed in the methods section.
Cat should be replaced by feline, line 32.
Author Response
Referee 1.
Jaroslav Flegr and colleagues submit for publication a study entitled “Cognitive Effects of Toxoplasma and CMV Infections: A Cross- 2 Sectional Study of 557 Young Adults Considering Modulation 3 by Sex and Rh Factor”.
It investigates an interesting aspect from an innovative angle by evaluating the associations between a selection of outcomes and two types of exposure, but calls for two major comments
(1) First, throughout the paper, reference is made to the "effects" in humans of toxoplasmosis or CMV infection. However, cross-sectional studies such as the current one, or those referred to in this document, are based on the concomitant evaluation of outcomes and exposures, with no follow-up, and do not allow conclusions to be drawn in terms of causality. This is a major point in terms of methodological and scientific rigor. This point is discussed in the discussion, and should concordantly be maintained throughout the manuscript. The numerous uses of the word "effect" should be replaced by "association".
We conducted a thorough review of the manuscript and made the necessary amendments throughout. In connection with this, we also excised the following technical note from the Methods section:
“We consistently use the term “effect” in its statistical sense, referring to the difference between the true population parameter and the null hypothesis value. Only in the Discussion section do we distinguish between cause and effect, using “effect” in its non-technical sense.”
(2) The second point concerns the assessment of exposure to toxoplasmosis or CMV infection, which is a major issue.
Concerning toxoplasmosis
(3) can the authors specify the cut off points for considering serological tests to be positive?
Please, see the item 9.
(4) If they are those recommended by the manufacturer, this should be indicated. If they have been modified, this should also be indicated and justified.
Please, see the item 9.
(5) How many patients were in doubtful zones for toxoplasmosis? How were they considered?
Please, see the item 6.
(6) The authors mention that participants with discordant results between the two tests for toxoplasmosis were excluded. How many participants were involved? At what point were they excluded?
The number of participants with discordant results fluctuated over time (surprisingly, it is usually higher in autumn). However, throughout the period of the study, these participants accounted for 5.27% of the total. These participants were excluded from the study before any data analysis was performed.
We included the following information in the Methods section of the corrected version of the manuscript:
“Subjects were considered non-infected with Toxoplasma if results from both tests were negative, while those with positive results in both tests were considered Toxoplasma positive. Approximately 5.3% of participants, who did not meet the condition of having either both negative or both positive results from two tests, were excluded from the study before any data analysis was performed.”
(7) It would not have been illogical to consider these patients as positive, in the context of a long standing infection profile and of IgG levels that had regressed and fluctuated around a threshold. To what extent would considering them as positive have altered the conclusions? This point needs to be discussed.
Most likely, some of the "negative" subjects are, in fact, infected with Toxoplasma. Unfortunately, these subjects are also, on average, infected for the longest time and therefore are most affected by the infection. In the past, we used to analyze data contaminated with potentially false negative subjects using a special randomization test (Flegr & Havlíček, Folia Parasitol., 46, 22–28, 1999). However, we no longer do so due to difficulties in publishing results based on such non-standard statistical tests. The censoring of subjects with low concentrations of anti-Toxoplasma antibodies has proven to be a safer approach. It probably results in a reduction of the observed strength of associations. However, the final results appear much more credible to readers, referees, and editors.
We included the following paragraph in the Limitation section:
“In the study, we did not include individuals with borderline concentrations of anti-CMV antibodies (0.36–0.44 IU/ml), and individuals whose results from CFT and ELISA tests for toxoplasmosis did not match. In the case of CMV, visual inspection of the distribution of antibody concentration values indicates that we likely only eliminated a small number of infected individuals, suggesting that this procedure had only a minimal impact on the obtained results. However, for toxoplasmosis, the boundary between the antibody concentration values of negative and positive individuals was much more blurred, and by filtering, we likely eliminated a much larger number of individuals infected with Toxoplasma. These were predominantly individuals who had been exposed to the infection for the longest period and thus were probably the most affected by the infection. Therefore, it can be assumed that the actual strengths of associations between latent toxoplasmosis and cognitive performance are higher in the general population than those measured in our censored sample. On the other hand, our many years of experience show that in some uninfected individuals, especially the levels of antibodies cross-reacting with Toxoplasma antigens in the CFT test are transiently or permanently elevated, likely due to ongoing inflammatory processes. These elevated levels of cross-reacting antibodies can also affect the results of ELISA tests. Including such falsely positive individuals in the study sample would likely significantly reduce the strength of associations between infection and cognitive performance, and additionally completely devalue the results of tests studying the correlations of antibody concentration with performance test outcomes.”
(8) How was antibody concentration defined, was it taken into account in a linear fashion, or were thresholds defined?
In the main parts of the study (results presented in Tables 2, 3, 4, and Fig. 1), we used the first approach. At first, we applied a threshold-based method to convert the results of diagnostic tests (continuous variables) into binary variables—“Toxoplasma-infected” and “CMV-infected.” These binary variables were then utilized to assess differences in cognitive performance between infected and non-infected subjects and in the path analyses. The rationale behind this approach is now elaborated upon in the newly added section on limitations within the discussion.
In the nonparametric correlation analyses, we utilized continuous variables (the concentration of anti-Toxoplasma or anti-CMV antibodies measured via ELISA) to ascertain the correlation between the concentration of antibodies and the outcomes of cognitive performance tests (results presented in Table 3, even columns). However, these analyses were conducted exclusively for the subpopulations of subjects infected with either Toxoplasma or CMV (independently), to prevent conflating data from subjects with specific antibody concentrations with those showing cross-reacting antibodies in non-infected or infected subjects.
We incorporated this information into the Materials and Methods section of the revised manuscript:
“At first, we applied a threshold-based method to convert the results of diagnostic tests (continuous variables) into binary variables—“Toxoplasma-infected” and “CMV-infected.” These binary variables were then utilized to assess differences in cognitive performance between infected and non-infected subjects and in the path analyses. In the correlation analyses, we utilized continuous variables (the concentration of anti-Toxoplasma or anti-CMV antibodies measured via ELISA) to ascertain the correlation between the concentration of antibodies and the outcomes of cognitive performance tests. However, these analyses were conducted exclusively for the subpopulations of subjects infected with either Toxoplasma or CMV (independently), to prevent conflating data from infected subjects with specific antibody concentrations with data from non-infected subjects with high levels of cross-reacting antibodies.”
(9) It should be clarified whether the cut-off values used were those of the manufacturer, or if not, how they were defined and why.
In the revised version of the manuscript, we explicitly stated that we used the cut-off values recommended by the manufacturers:
“Toxoplasma infection status was assessed at the National Reference Laboratory for Toxoplasmosis, National Institute of Public Health in Prague. We employed the homemade complement fixation test (CFT) using TestLine Clinical Diagnostic antigen and ELISA IgG test (TestLine Clinical Diagnostics) for this purpose. ELISA assay cut-point values were established according to the manufacturer’s instructions.
CMV status (infected vs. non-infected) was determined at the National Reference Laboratory for Herpes Viruses, National Institute of Public Health in Prague using the sandwich ELISA method with an inactivated CMV AD 169 strain antigen (ETI CYTOK G PLUS, DiaSorin, Saluggia, Italy), allowing quantitative detection of CMV IgG from 0.4 IU/ml to 10 IU/ml. We used the threshold recommended by the manufacturer, however, we excluded about 2.8% subjects with ambiguous serological test results (IgG concentration between 0.36–0.44 IU/ml) from the study.”
(10) Why were patients with low values excluded? It would have been interesting to analyze the data by considering them as weakly positive, to see the impact on the results.
In principle, the exclusion of patients with low values is due to our inability to distinguish between subjects with low concentrations of specific anti-Toxoplasma antibodies and those with high concentrations of cross-reacting antibodies in routine diagnostic tests. For a more detailed explanation, please refer to our response to item (7).
(11) How was antibody concentration defined, was it taken into account in a linearly, or were thresholds defined?
As we explained in more detail above (8) we used both methods, each of them for different purposes.
(12) The authors should also comment in the discussion on the difficulty of estimating the reality of exposure to toxoplasmosis and CMV on the basis of a single sample. The entire study is based on the analysis of a single sample per patient. This is an major intrinsic limitation of this type of survey, which must be discussed, particularly as regards the lack of sensitivity of the tests used.
The standard method for diagnosing Toxoplasma and CMV infections involves the examination of only one sample, not only for research purposes but also for clinical ones. In the case of examining pregnant women, the test is repeated during pregnancy, not because the tests have low sensitivity or specificity, but to timely detect infections that occurred during pregnancy, which could cause congenital toxoplasmosis. Naturally, no test has 100% specificity and 100% sensitivity. Therefore, in our sample, there might be a few falsely positive individuals and a higher number of falsely negative individuals. Such stochastic errors can pose a fatal problem in clinical practice, and it is necessary to minimize their occurrence by all means. Generally, however, they are not considered a fundamental problem in research projects. In realistic scenarios, they can only lead to a falsely negative result of the study—failing to demonstrate an existing association (Type II error), but not to a falsely positive test result—proving a non-existent association (Type I error).
We included the following comments in the section Limitations:
“Similarly, our data set must contain some false negative individuals, i.e., individuals who tested negative in the serological test but were in fact infected. The serological tests used are considered reliable and are utilized not only for research purposes but also in clinical practice. Reference laboratories at the National Health Institute represent top diagnostic facilities for toxoplasmosis and herpesviruses, setting (and overseeing) the standard for other diagnostic facilities in the Czech Republic. Even so, it is certain that at least some of the individuals who tested negative are in fact infected; due to the long duration from infection to testing, their antibody levels have fallen below the threshold value. Again, such an error could cause a Type II error (failing to identify weak effects), but it would not elevate the risk of a Type I error (identifying non-existent effects)【74].”
In addition
(13) The introduction could be shortened, and certain elements included in the discussion.
We shortened some parts of the introduction, especially by radically reducing expendable passages related to Rh polymorphism. Please, check the version of the manuscript with highlighted changes.
(14) The results section contains many elements that should be placed in the methods section.
We carefully reviewed the section for such elements and eliminated those parts where it did not jeopardize the comprehensibility of the text, e.g., lines 340-347, from the Results section.
(15) Cat should be replaced by feline, line 32.
Done. Thank you for your time and help
Reviewer 2 Report
Comments and Suggestions for Authors
The Authors of the publication titled 'Cognitive Effects of Toxoplasma and CMV Infections: A Cross-Sectional Study of 557 Young Adults Considering Modulation by Sex and Rh Factor' present very interesting results.
The research was well planned and described. The description of the results is clear, but the analysis of the tables poses many difficulties due to the number of analyzes and factors examined. Perhaps it would be better to present the results in the form of graphs with significance statistics.
In several places in the text, Toxoplasma is written without italics, e.g. line 441, 442. Please make sure vs., via and other Latin phrases should not be written in italics either.
The discussion is well-written and refers definitively to the results obtained during the research. I have only one reservation to the discussion, I do not agree with the opinion that based on the IgG antibody titer we can talk about the phase of toxoplasmosis. Without determining at least the avidity of IgG antibodies, I would not draw such far-reaching conclusions. It would be best to perform a whole panel of tests, i.e. determine IgM and IgG antibodies and the avidity of IgG antibodies. Only then we will correctly determine the phase of the disease. Having a well-defined early/acute or chronic/latent phase of the disease, it may turn out that there are significant differences between them. In my work, I have often encountered cases where the titer of IgG antibodies did not indicate the phase of the disease, even when samples were taken at intervals. In some patients the titer may be low all the time, in others, it may be very high even for several years. As well as long-lasting IgM in some patients. Please change this in the discussion.
Comments on the Quality of English Language
I have no objections, the work is well-written.
Author Response
Referee 2
The Authors of the publication titled 'Cognitive Effects of Toxoplasma and CMV Infections: A Cross-Sectional Study of 557 Young Adults Considering Modulation by Sex and Rh Factor' present very interesting results.
Thank you.
(16) The research was well planned and described. The description of the results is clear, but the analysis of the tables poses many difficulties due to the number of analyzes and factors examined. Perhaps it would be better to present the results in the form of graphs with significance statistics.
We agree that the number of hypotheses tested in this study is high. In our opinion, the only way to present such a quantity of results is through tables. We acknowledge that presenting data in graphs is usually more illustrative than in tables. Unfortunately, it is also true that the number of graphs needed to present all the results would be unbearably high. If we were publishing in a print journal, we might consider moving some of the four tables to the supplement. However, in the case of electronic journals, such a move solves nothing and only worsens the convenience for readers. Therefore, we prefer to keep the tables as integral parts of the article.
(17) In several places in the text, Toxoplasma is written without italics, e.g. line 441, 442. Please make sure vs., via and other Latin phrases should not be written in italics either.
Thank you for pointing out this problem. We have carefully reviewed the new version of the article and added italics to all Latin names and phrases.
(18) The discussion is well-written and refers definitively to the results obtained during the research. I have only one reservation to the discussion, I do not agree with the opinion that based on the IgG antibody titer we can talk about the phase of toxoplasmosis. Without determining at least the avidity of IgG antibodies, I would not draw such far-reaching conclusions. It would be best to perform a whole panel of tests, i.e. determine IgM and IgG antibodies and the avidity of IgG antibodies. Only then we will correctly determine the phase of the disease. Having a well-defined early/acute or chronic/latent phase of the disease, it may turn out that there are significant differences between them. In my work, I have often encountered cases where the titer of IgG antibodies did not indicate the phase of the disease, even when samples were taken at intervals. In some patients the titer may be low all the time, in others, it may be very high even for several years. As well as long-lasting IgM in some patients. Please change this in the discussion.
We agree that the concentration of antibodies cannot be used to estimate the length of infection on an individual basis. For clinical purposes, such as distinguishing between old and new infections in pregnant women, this method is utterly unsuitable, and it is always necessary to rely on antibody avidity, the dynamics of changes in antibody concentration measured during repeated sample collections, and also on clinical data. Statistically, however, it roughly holds true that individuals who have been infected recently (but, of course, not very recently) have lower levels of IgG antibodies than those infected a long time ago. Therefore, when studying the correlation of the intensity of phenotypic changes associated with infection with the length of infection on large samples of infected individuals, it is possible to use the level of IgG antibodies as a crude proxy for the length of infection.
We are currently completing a study in which we have repeatedly examined the levels of anti-Toxoplasma IgG antibodies in several tens of thousands of professional soldiers over the course of twenty years. It appears that for many of them, the levels do not change much, for many others randomly decrease and then increase again. However, for the majority of individuals, there is a clear trend of a gradual decrease in antibody levels, and for many participants, the antibody level even dropped temporarily or permanently below the positivity threshold over the years.
We included the following paragraph in the Limitations section:
"In the present study, we used the concentration of specific IgG antibodies as a proxy for the duration of infection. We would like to remind readers that this proxy is relatively rough, as this concentration may fluctuate or remain relatively stable in many subjects over time. Generally, however, after a rapid increase during the first 1-2 months post-infection, the concentration of IgG antibodies in most individuals decreases over the duration of latent toxoplasmosis. Therefore, in research studies conducted on sufficiently large samples, the level of IgG antibodies can be used to estimate the duration of latent toxoplasmosis. It is important to emphasize that the level of these antibodies cannot be used to estimate the duration of infection in individuals, for example, in pregnant women, where entirely different methods based on measurements over time, on antibody avidity measurements, and on monitoring clinical symptoms must be used."
Comments on the Quality of English Language
I have no objections, the work is well-written.
Thank you.
Reviewer 3 Report
Comments and Suggestions for Authors
Cognitive Effects of Toxoplasma and CMV Infections: A Cross-Sectional Study of 557 Young Adults Considering Modulation by Sex and Rh Factor
The authors report that Toxoplasma infections with noumrous effects on human cognitive performance. Similar for CMV and humans.
It is an interesting paper.
I recommend for the introduction and discussion part to take in consideration the transmission of Tg via water and food and cite related papers.
Abstract
Too long, lines 11-15 not needed.
Abstract edited version
One-third of humanity harbours a lifelong infection with Toxoplasma gondii. This parasite undergoes sexual reproduction in cats and asexual reproduction in any warm-blooded intermediate hosts. The cycle progresses as cats ingest these hosts, containing the parasite's tissue cysts. Such infections can alter behaviours in animals and humans, potentially increasing predation risk by felines—usually seen as parasite-induced manipulations. This study aims to delineate toxoplasmosis's effects on cognitive abilities and compare these to the effects of human cytomegalovirus (CMV), which also infects the brain but is not spread through predation. We evaluated the cognitive performance of 557 students who had been examined for Toxoplasma and CMV infections, using intelligence, memory, and psychomotor tests. Results indicated cognitive impairments in seropositive in individuals for both pathogens, with variations in cognitive impact related to sex and Rh factor. Specifically, Toxoplasma was associated with lower IQ in men, whereas CMV predominantly had worse 21 women's memory and reaction speeds. Analysis of antibody concentrations hinted that certain Toxoplasma was associated cognitive detriments might wane (impaired intelligence) or worsen (impaired reaction times) over time following infection. The findings imply that cognitive impairments from neurotropic pathogens are likely due to pathological changes in the brain rather than direct manipulative actions by the parasites.
I recommend linguistic improvements for more clarity and transparency and errors.
Whole MS
l.17, pl use cursive for T
References
Please check one by one and correct based on the Journals style. There irregularities in the citation style.
Author Response
Referee 3
The authors report that Toxoplasma infections with noumrous effects on human cognitive performance. Similar for CMV and humans.
It is an interesting paper.
Thank you.
(19) I recommend for the introduction and discussion part to take in consideration the transmission of Tg via water and food and cite related papers.
Thank you for this recommendation, which would have provided a pretext to add a paragraph on the transmission of toxoplasmosis. Including a mention of waterborne transmission would mean adding an entire new section about many important pathways of transmission, likely including the sexual transmission route that we recently described in our papers. Although this route is not as significant from an epidemiological standpoint compared to water transmission, it could be exceptionally important from a clinical perspective as it may disproportionately often lead to the outbreak of congenital toxoplasmosis. Ultimately, we resisted the temptation and did not include a section on the transmission of toxoplasmosis in the article, as it is only tangentially related to its theme and as it goes against the recommendation of Referee 1, who asked to shorten the Introduction section instead. However, the very first citation in our article is an excellent review by Tenter et al., in which the pathways of toxoplasmosis transmission are described in great detail.
Abstract
Too long, lines 11-15 not needed.
We abbreviated the critical part of the Abstract from:
One-third of humanity harbors a lifelong infection with Toxoplasma gondii. This parasite undergoes sexual reproduction in cats and asexual reproduction in any warm-blooded intermediate hosts. The cycle progresses as cats ingest these hosts, containing the parasite's tissue cysts. Such infections can alter behaviors in both animals and humans, potentially increasing predation risk by felines—usually seen as parasite-induced manipulations. This study aims to delineate toxoplasmosis's effects on cognitive abilities and compare these to the effects of human cytomegalovirus (CMV), which also infects the brain but is not spread through predation.
to
One-third of humanity harbors a lifelong infection with Toxoplasma gondii, and probably about 80% are infected with human cytomegalovirus (CMV). This study aims to delineate the associations between toxoplasmosis and cognitive abilities and compare these to the associations with CMV.
(21) I recommend linguistic improvements for more clarity and transparency and errors.
We double-checked the manuscript for transparency and errors. Please refer to the manuscript with highlighted changes for the modifications we made.
Whole MS
(22) l.17, pl use cursive for T
Thank you for pointing out this problem. We have carefully reviewed the new version of the article and added italics to all Latin names and phrases.
(23) References
Please check one by one and correct based on the Journals style. There irregularities in the citation style.
We reviewed the References section and standardized all entries to conform to the journal's style.
Round 2
Reviewer 3 Report
Comments and Suggestions for Authors
The new version has been improved. However, what I am missing in this paper is the transmission pathways of the Tg, which are waterborne, foodborne, and airborne. Please use original literature from recent decades reporting the detection of Tg in water, food, and air. These are new aspects, and the related literature with related information in the introduction and the discussion should be considered. It is essential to include these aspects in your paper.
Comments on the Quality of English Languagesee review
Author Response
We have added a new figure to the manuscript with a diagram showing possible transmission pathways of toxoplasmosis. In the legend, we described newly identified pathways (including airborne transmission) and added the corresponding citations to the literature list:
Figure 1. Transmission of Toxoplasma gondii through old and recently described pathways. A) Environmental contamination with oocysts affects soil, water, plant life, and potentially air. B) Small animals contract toxoplasmosis through ingesting contaminated food and environmental exposure. C) Predation leads to the transfer of the parasite as definitive hosts consume infected prey. D) Livestock become infected via contaminated feed and environmental sources, serving as intermediate hosts. E) Humans risk infection by consuming undercooked meat harboring tissue cysts and unpasteurized dairy products with tachyzoites. F) Human infection can also occur from contaminated vegetables, water, or other foods, such as fish from contaminated sources, with potential direct infection from contact with feline feces. G) Airborne transmission might be possible for humans and animals in highly contaminated environments [2]. H) Sexual transmission in humans, specifically through penile-vaginal [3] or potentially oral routes [4], can be significant due to the risk of congenital toxoplasmosis. I) Transmission through blood transfusion or organ transplantation is an acknowledged risk, albeit with stringent screening processes in place. J) Congenital toxoplasmosis results from maternal-fetal transmission and represents a significant route of infection with implications for neonatal health.